# Evaluation of Structure and Corrosion Behavior of FeAl Alloy after Crystallization, Hot Extrusion and Hot Rolling

**DOI:** 10.3390/ma13092041

**Published:** 2020-04-27

**Authors:** Janusz Cebulski, Dorota Pasek, Bartosz Chmiela, Magdalena Popczyk, Andrzej Szymon Swinarew, Arkadiusz Stanula, Zbigniew Waśkiewicz, Beat Knechtle

**Affiliations:** 1Faculty of Materials Engineering, Silesian University of Technology, 40-019 Katowice, Poland; janusz.cebulski@polsl.pl (J.C.); dorota.pasek@polsl.pl (D.P.); bartosz.chmiela@polsl.pl (B.C.); 2Faculty of Science and Technology, University of Silesia in Katowice, 41-500 Chorzów, Poland; magdalena.popczyk@us.edu.pl (M.P.); andrzej.swinarew@us.edu.pl (A.S.S.); 3Institute of Sport Science, The Jerzy Kukuczka Academy of Physical Education, 40-065 Katowice, Poland; a.stanula@awf.katowice.pl (A.S.); z.waskiewicz@awf.katowice.pl (Z.W.); 4Department of Sports Medicine and Medical Rehabilitation, Sechenov University, 119991 Moscow, Russia; 5Institute of Primary Care, University of Zurich, 8091 Zurich, Switzerland

**Keywords:** intermetallic phase, FeAl, electrochemical corrosion

## Abstract

The paper presents the results of tests on the corrosion resistance of Fe40Al5Cr0.2TiB alloy after casting, plastic working using extrusion and rolling methods. Examination of the microstructure of the Fe40Al5Cr0.2TiB alloy after casting and after plastic working was performed on an Olympus GX51 light microscope. The stereological relationships of the alloy microstructure in the state after crystallization and after plastic working were determined. The quantitative analysis of the structure was conducted after testing with the EBSD INCA HKL detector and the Nordlys II analysis system (Channel 5), which was equipped with the Hitachi S-3400N microscope. Structure tests and corrosion tests were performed on tests cut perpendicular to the ingot axis, extrusion direction, and rolling direction. As a result of the tests, it was found that the crystallized alloy has better corrosion resistance than plastically processed material. Plastic working increases the intensity of the electrochemical corrosion of the examined alloy. It was found that as-cast alloy is the most resistant to corrosion in a 5% NaCl compared with the alloys after hot extrusion and after hot rolling. The parameters in this study show the smallest value of the corrosion current density and corrosion rate as well as the more positive value of corrosion potential.

## 1. Introduction

The intensive development of material engineering in recent years has allowed the development and production of innovative alloys based on the FeAl intermetallic phase. FeAl-based alloys are of particular interest due to the Al content of 40% at., and thus the density of 5.4–6.7 g·cm^−3^ and the relatively low price of input materials, compared to the price of alloying elements of heat-resistant steels, containing chromium, nickel, and molybdenum [1]. Material with a structure with the dominant share of ordered intermetallic phases from the Fe-Al system has properties that allow it to be used as a structural material for elements operating at elevated temperatures, often in an aggressive environment, as well as in one or many oxidants [2,3,4]. Research conducted by Kulak and Kupka [5] and other researchers [6,7] shows that resistance results from the formation of a passive aluminum oxide layer on the surface of the material. Additionally, alloys based on the FeAl intermetallic phase are characterized by resistance to abrasive, erosive and cavitation wear [8,9]. In order to improve their plasticity as well as their resistance to brittle cracking, they are subjected to plastic working processes, obtaining a fine-grained structure. The alloy structure belongs to the factors determining its corrosion resistance [4]. The results of research conducted in the last two decades indicate that grain refinement has a positive effect on both the plasticity and strength of FeAl alloys [10,11]. The wide spectrum of properties of alloys based on the intermetallic phase of FeAl allowed the application of these materials in many industries, mainly due to their heat resistance and high resistance to erosion. For this reason, the greatest use of FeAl alloys found in the energy industry is for elements of boilers, burners, gas filters, heat exchangers and pipes. In the chemical industry, they are used for tools, pipes and containers. They are also used in the petrochemical, food, and automotive industries [12,13,14]. Although they are particularly advantageous for high-temperature service, these materials could be also suitable for room-temperature applications [15,16]. For such applications, an understanding of the aqueous corrosion behavior is of great importance. The reason for conducting tests in the field of corrosion resistance in a liquid environment of heat-resistant alloys on the matrix of the FeAl intermetallic phase is the fact that structural components operating at high temperatures are exposed to the effects of gases containing acid anhydrides. While the phenomenon of electrochemical corrosion does not occur during operation at high temperatures, when these devices are turned off, and consequently, when the temperature drops to room temperature, the water contained in the air (water vapor) may combine with the anhydrides mentioned, resulting in the formation of aqueous acid or saline solution. In addition, corrosion resistance tests on FeAl intermetallic matrix alloys are usually conducted on the material after casting [17], because these materials are characterized by brittleness at ambient temperature [18] and belong to the group of hard-deformable materials [19] or for the Fe_3_Al phase [20,21]. The chemical composition of the tested alloy is also important. The research conducted in this work was based on a 40% alloy content aluminum. The Fe-Al phase equilibrium system developed by Kubaszewski [22] shows that, at this content, the alloy remains single-phase throughout the entire solid-state range. In addition, numerous studies conducted by Barcik and Cebulski [10,23] indicate that an increase in the proportion of aluminum causes a decrease in the plasticity of the alloy, which in turn hinders its plastic processing. Lowering the alloy content below 36% aluminum causes the Fe_3_Al phase to occur. In addition, studies conducted by McKamey, Klein, and others [24,25] prove that the addition of chromium to FeAl alloy in an amount of about 2% to 6% increases the plasticity of Fe-Al alloys, thus limiting the phenomenon of hydrogen embrittlement. Research carried out in [26,27] presents a summary of the impact of grain size on the rate of electrochemical corrosion for various materials and environments. These analyses do not include the presentation of corrosion resistance for the intermetallic alloy Fe40Al5Cr0.2TiB, therefore, in order to determine the correlation between the results for different materials, additional tests and analyses of their results should be performed for the tested alloy. This is due to the fact that, in particular, the intermetallic alloy has not been characterized in previous studies in a way that gives grounds to accept representative results from a statistical point of view.

So far, the influence of alloy plastic working on the matrix of the FeAl intermetallic phase with a 40% content has not been studied, ith micro-additives for corrosion resistance in an NaCl environment. A 5% NaCl solution was used to obtain standardized results.

Given the above purposeful information, the investigation aims to study the room-temperature corrosion behavior in a chloride-containing solution alloy Fe40Al5Cr0.2TiB after casting and plastic processing (hot extrusion, hot rolling).

## 2. Materials

The tests were carried out on samples made of Fe40Al5Cr0.2TiB intermetallic alloy (chemical composition is shown in Table 1). For melting, ARMCO iron (technically pure) (Katowice, Poland), ARO aluminum (99.995% by mass) (Katowice, Poland), aluminothermic chromium (Katowice, Poland) obtained using the Kroll method and amorphous boron (Katowice, Poland) (technically pure) were used. The smelting was carried out in a Balzers VSG-2 vacuum induction furnace (Balzers, Liechtenstein). The basic components in the form of iron and aluminum were placed in an alundum crucible before the melting process began while alloying additives were introduced into the metal bath in the order of their increasing reactivity. Smelting was carried out in vacuo. The metal bath was heated to 1500 °C and cast into a mold in the atmosphere of the furnace in which the melt was carried out.

Because the material after casting was characterized by the fragility and coarseness of the microstructure and the heterogeneity of its chemical composition, heat treatment was applied. Homogenizing annealing was carried out at a temperature of 1050 °C for 72 h, homogenizing the chemical composition of the alloy. The next stage was plastic working by the method of concurrent extrusion and rolling in sheaths. The extrusion process was carried out in the manner that is the subject of patent No. 208310 [11]. The application of this method, in comparison to extrusion in a conventional manner, allows obtaining plastic material without cracks, thereby improving the plastic properties of the alloy as well as homogeneity of the microstructure and grain refinement. The input material was a cast made of Fe40Al5Cr0.2TiB alloy in the shape of a cylinder with a diameter of 22.2 mm, and after extrusion, the diameter of the material was 15.7 mm. Rolling of the Fe40Al5Cr0.2TiB alloy was carried out under the industrial conditions of the Baildon Steelworks (Katowice, Poland) on a furrow mill. Cylindrical ingots with a diameter of 25 mm and a length of 300 mm, after homogenization at a temperature of 1050 °C for 72 h, were forged into pipes made of austenitic steel X5CrNi18–10 with a wall thickness of 3 mm. Ingots in casings were annealed in a gas-fired industrial furnace to a temperature of about 1250 °C for 0.5–0.75 h and subjected to rolling. Plastic working by rolling was carried out in four culverts. After each pass, the rolling stock was heated to the assumed temperature of 1250 °C. From the original dimensions of the ingots after rolling, rods with a diameter of about 7 mm were obtained (Table 2).

## 3. Methods

The research aimed to determine the corrosion resistance of Fe40Al5Cr0.2TiB alloy with different microstructures after crystallization and shaped during plastic processing in a 5% NaCl environment. The research program included:Analysis of the alloy microstructure after casting and after plastic working (rolling, extrusion).Quantitative analysis of the Fe40Al5Cr0.2TiB alloy microstructure after individual technological stages.Corrosion resistance of all FeAl alloys was determined, using the potentiodynamic polarization technique. These measurements were conducted in the 5% NaCl solution, using three-electrode cell and PGSTAT30 Potentiostat / Galvanostat electrochemical system (Metrohm Autolab B.V., Utrecht, Netherlands). The reference electrode was a saturated calomel electrode (SCE) and a counter electrode was a platinum mesh. A geometric surface area of all tested alloys was equal to 1 cm^2^. Potentiodynamic curves were registered in the potential range ± 250 mV relative to the open circuit potential with rate *v* = 1 mV·s^−1^.Tests of the surface condition after corrosion tests along with X-ray microanalysis of EDS chemical composition.

Examination of the microstructure of the Fe40Al5Cr0.2TiB alloy after casting and after plastic working was performed on an Olympus GX51 light microscope (Tokyo, Japan).

The stereological relationships of the alloy microstructure in the state after crystallization and after plastic working were determined. The quantitative analysis of the structure was made after testing with the EBSD INCA HKL detector (HKL Technology, Hobro, Denmark) and the Nordlys II analysis system (Channel 5), which was equipped with the (Hitachi S-3400N microscope). The analysis of grain size, grain orientation and texture was performed. Orientation maps were displayed in the inverse pole figures color scheme (which allows the crystallographic orientation to be quickly interpreted in terms of the sample coordinate system). Texture analysis was carried out based on pole figures (PFs) (revealing how plane normals are arranged relative to the specimen) and the inverse pole figures (IPFs) (revealing which crystallographic directions align with the specimen axes).

The surface appearance of the samples after corrosion tests was studied by Scanning Electron Microscopy (SEM) (Hitachi S-4200, Tokyo, Japan). The chemical composition was tested using an X-ray energy dispersion spectrometer (EDS) from ThermoNoran (System Seven) (Waltham, MA, United States) at a voltage accelerating the electron beam of 15 keV. The spectrometer was connected to the cited microscope.

Structure tests and corrosion tests were performed on tests cut perpendicular to the ingot axis, extrusion direction, rolling direction.

## 4. Results and Discussion

### 4.1. Microstructure Research

The alloy based on the intermetallic phase Fe40Al5Cr0.2TiB after casting is characterized by a heterogeneous and coarse-grained microstructure. The coarseness of the alloy after casting and high resistance to shaping accompanying hot plastic processing make the material deform unevenly. This results in the occurrence of areas of varying grain size on the cross-section of the deformed material. The extrusion process allows obtaining a fragmented microstructure of the examined alloy. Plastic working by rolling allows for obtaining a homogeneous, fine-grained microstructure (Figure 1). However, this technology is associated with the use of shields that complicate the process and increase the cost of manufacturing the blank.

### 4.2. Quantitative Microstructure Analysis

Microstructural investigations of FeAl alloy by EBSD technique revealed differences in grain size and grain orientation in as-cast state, after hot extrusion and after hot rolling. The as-cast alloy was characterized by a typical primary structure with columnar grains in the outer zone and equiaxed grains in the middle part. The grains were characterized by a diversified crystallographic orientation with high angle boundaries (HABs, misorientation higher than 15°), which accounted for 92% of all grain boundaries in the investigated area. Some low angle boundaries (LABs, misorientation lower than 15°) were found (Figure 2).

After hot extrusion, the grain orientation diversity was different. Moreover, the grains were characterized by a subgrain structure with many LABs, which accounted for 84% of all grain boundaries in the investigated area (Figure 3).

The grain orientation after hot rolling was similar to the as-cast alloy—there were mainly HABs (75% of all grain boundaries)—as shown in Figure 4.

Evaluation of grain size (understood as planar section area of a grain) revealed that grain size distributions were similar in the case of the as-cast alloy and the extruded bar. In the case of as-cast alloy, the finest grains with a size within a 0–0.05 mm^2^ interval constitute 70% of all grains in the investigated area, but in the case of the extruded bar it was 75%. After hot extrusion, few grains bigger than in as-cast alloy were found, which testifies to the grain coarsening during the hot extrusion. However, after hot rolling the strong grain refinement was observed (Figure 5).

Grain parameters in the FeAl alloy are shown in Table 3. The average grain size after hot extrusion was bigger in comparison to as-cast alloy and was characterized by a variation coefficient almost 50% higher, which indicates a higher diversification of grain size than in case of as-cast alloy. After hot rolling the average grain size was much lower.

The texture analysis of the FeAl alloy was performed to confirm the preferred crystallographic orientations. Texture investigations were carried out on the cross-sections of as-cast alloy, extruded bar (perpendicular to the extrusion axis) and the hot rolled sample (perpendicular to the rolling direction). No texture was found in the as-cast alloy; on the pole figures (PFs) the poles of {110} and {111} planes were distributed uniformly (Figure 6). The poles visible on the inverse pole figures (IPFs) reveal that the parallelism of <110> and <001> directions to the orthogonal transverse directions (TD_1_ and TD_2_) was random (in the alloy after solidification there are no factors forcing the specified orientations).

However, in the alloy after hot extrusion, the fiber texture was found. The theoretical and experimental {110} and {111} PFs and IPFs for transverse directions and normal direction presenting the <110> and <001> fiber textures are shown in Figure 7. The theoretical PFs and IPFs take into account all possible unit cell rotations around <110> and <001> directions. A comparison of theoretical and experimental PFs and IPFs confirmed the presence of <110> fiber texture (typical for the metals and alloys with A2 and B2 lattices), characterized by the parallelism of <110> direction and the extrusion direction.

The alloy after hot rolling was characterized by the {111} <uvw> texture; the {111} planes were parallel to the rolling plane, but there were no strongly preferred crystallographic directions parallel to the rolling direction (Figure 8). A comparison between the theoretical and experimental PFs and IPFs revealed that the ideal orientations determined were the {111} <110> and {111} <112> textures. In both cases, the {111} plane was parallel to the rolling plane and <110> and <112> directions were parallel to the rolling direction (Figure 8).

### 4.3. Corrosion Resistance Tests

Open circuit potentials of all alloys were determined for 24 h (Figure 9). These investigations were carried out in order to stabilize the potential of the tested alloys in NaCl solution. The stabilized value of this parameter can be treated as an approximate value of the corrosion potential. The very similar nature of the *E* = f(*t*) relationship can be observed for FeAl alloy as-cast and after hot rolling. For all tested alloys, the potential stabilized after about 8 h. In the case of alloy after hot extrusion, an increase in the direction of more positive potentials was observed, which at first might have suggested the creation of passive protective layers, however, after 8 h there was a quite rapid decrease leading to the stabilization of potential. A range of ± 250 mV was chosen from the determined, stable value of open circuit potential and a potentiodynamic curve was recorded for all alloys (Figure 10). In this figure, experimental Tafel plots are presented using points. Solid lines denotes the data fitted using the Butler–Volmer equation: *j* = *j*_corr_ {exp [2.303(*E* − *E*_corr_) / *b**_a_*] − exp [−2.303(*E* − *E*_corr_)/*b**_c_*]} where *j* is current density (in A·cm^−2^) and *E* is potential (in V). On this basis, the values of corrosion parameters, i.e., corrosion potential (*E*_corr_), corrosion current density (*j*_corr_), anodic Tafel slope (*b*_a_), cathodic Tafel slope (*b*_c_) and polarization resistance (*R*_p_) for all alloys were determined (Table 4). Corrosion rate was calculated based on the following formula: *V*_corr_ [in mm·year^−1^] = 3.27 *j*_corr_ [in mA·cm^−2^] (*EW*/*d*), where *EW* is equivalent weight and *d* is density.

It was ascertained for all tested alloys, that anodic Tafel slopes have lower values than cathodic Tafel slopes, which means that the kinetics of the cathode process is faster compared to the anodic reaction rate. The corrosion potential for the as-cast alloy is more positive compared to the corresponding values of the alloys after hot extrusion and after hot rolling (Figure 10, Table 4). It was also noted that for the as-cast alloy, the value of corrosion current density is lowest compared to the other alloys (Table 4). This suggests that, the as-cast alloy exhibits the best anticorrosion properties in a 5% NaCl solution compared with the alloys after hot extrusion and after hot rolling. The proof of this is also the highest value of polarization resistance (about 14 kΩ·cm^2^) and lowest value of corrosion rate (only 0.028 mm·yr^−1^—which means, that corrosion processes occur with the lowest intensity).

### 4.4. Research Using a Scanning Electron Microscope

Tests on the surface condition of the Fe40Al5Cr0.2TiB alloy after corrosion tests in NaCl solution showed the presence of pits on the surface of all tested samples. It was observed that for the material after casting the number of localized attacks on the surface was smaller than for the wrought material. The largest surface damage after corrosion tests (number and size of pits) occurred for the material after the rolling process. Figure 11 shows the results of observation of the surface of the Fe40Al5Cr0.2TiB alloy after casting, which shows the effect of surface digestion, and the pits ran along the grain boundaries. It was found that pits developed primarily deeper into the material, which in turn may cause whole grains to fall out. Figure 12 and Figure 13 present the results of observing the surface of the Fe40Al5Cr0.2TiB alloy after plastic working (extrusion and rolling). Large, deep pitting spots were present on the surface. In addition, pits occurring on the surface of the material after rolling had the shape of gutters.

No corrosion products or passive layers formed on the surface of the samples after corrosion tests were performed.

EDS microanalysis of the chemical composition of the sample surface after corrosion tests showed the presence of elements included in the alloy.

Phenomena on the surface of tested materials are associated with selective corrosion. The diversity of topography of samples after corrosion tests, in particular pitting, may be associated with the crystallographic orientation of grains. The place of initiation of corrosive processes in individual grains are grain boundaries, as defective areas, and therefore more susceptible to the initiation of physico-chemical processes. The tested material is single-phase, therefore differences in phenomena occurring in the material may result from the structure (grain size, crystallographic orientation, defects of metallurgical origin). Depending on the degree of plastic working of the material, there is a different share of grain boundaries in the material. Plastic working causes a metastable state of the material, which may cause greater susceptibility to corrosion.

## 5. Conclusions

It was found that as-cast alloy is the most resistant to corrosion in a 5% NaCl compared with the alloys after hot extrusion and after hot rolling. The parameters in this study showed the lowest value of the corrosion current density and corrosion rate, the highest value of the polarization resistance as well as the more positive value of corrosion potential. The results obtained clearly indicate the accuracy of the selection of tests carried out; the analytical methods used show the corrosion effect consistent with theoretical assumptions. The results obtained are significantly influenced by the size and distribution of grains, as well as the boundaries between the grains that run uniformly without causing additional foci. Corrosion develops faster with a smaller grain system due to the larger ratio of grain boundary surfaces to the real grain surface, which increases the contact surface and accelerates corrosion.

## Figures and Tables

**Figure 1 materials-13-02041-f001:**
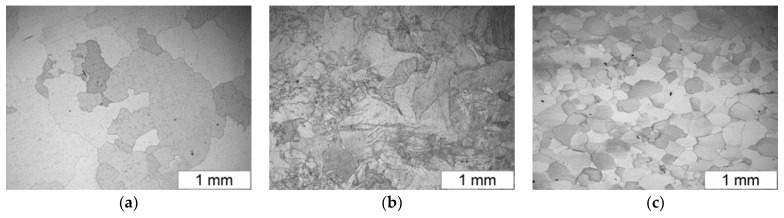
The microstructure of the Fe40Al5Cr0.2TiB alloy: (**a**) after casting, (**b**) after hot extrusion, (**c**) after hot rolling.

**Figure 2 materials-13-02041-f002:**
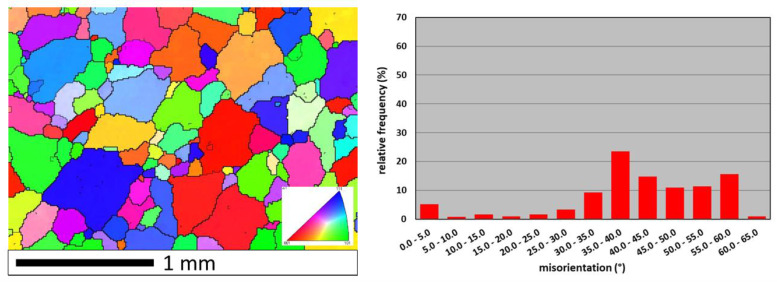
Orientation map and misorientation distribution for FeAl as-cast alloy: thick lines—high angle boundaries (HABs) (misorientation > 15°); thin lines—low angle boundaries (LABs) (misorientation < 15°); color interpretation according to the stereographic triangle (red <001> direction, green <110> direction, blue <111> direction).

**Figure 3 materials-13-02041-f003:**
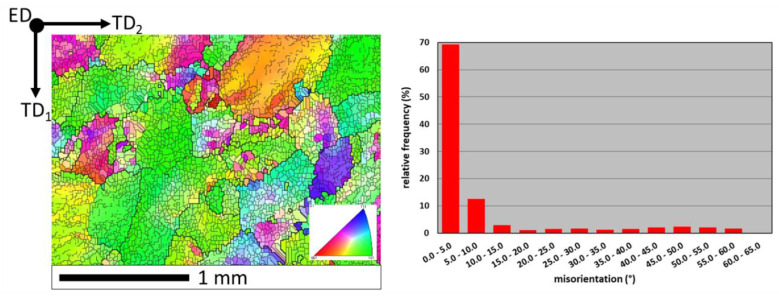
Orientation map and misorientation distribution for FeAl alloy after hot extrusion: thick lines—HABs (misorientation > 15°); thin lines—LABs (misorientation < 15°); color interpretation according to the stereographic triangle (red <001> direction, green <110> direction, blue <111> direction); TD_1_, TD_2_—orthogonal transverse directions; ED—extrusion direction.

**Figure 4 materials-13-02041-f004:**
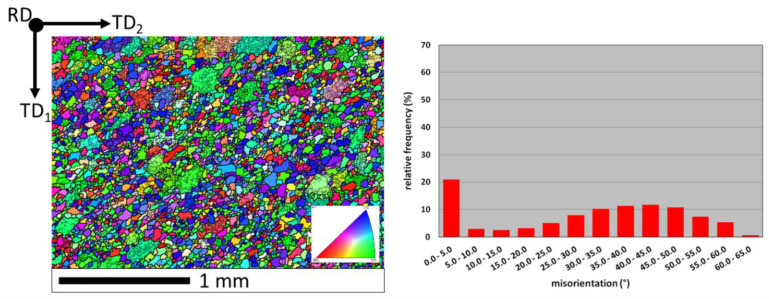
Orientation map and misorientation distribution for FeAl alloy after hot rolling: thick lines—HABs (misorientation > 15°); thin lines—LABs (misorientation < 15°); color interpretation according to the stereographic triangle (red <001> direction, green <110> direction, blue <111> direction); TD_1_, TD_2_—orthogonal transverse directions; RD—rolling direction.

**Figure 5 materials-13-02041-f005:**
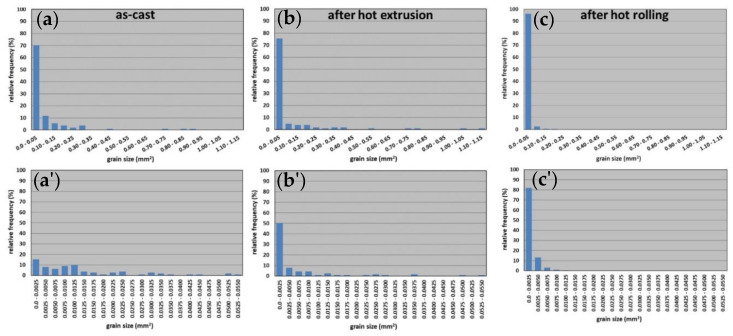
Grain size distribution for the FeAl alloy, (**a**) (**a’**) as-cast, (**b**) (**b’**) after hot extrusion, (**c**) (**c’**) after hot rolling.

**Figure 6 materials-13-02041-f006:**
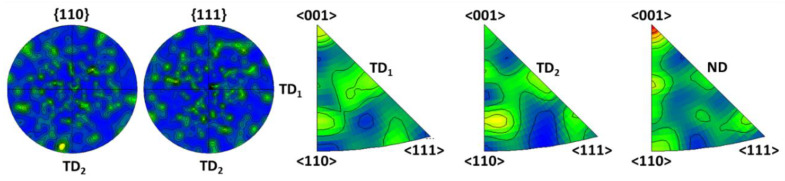
Pole figures (PFs) and inverse pole figures (IPFs) for the FeAl as-cast alloy: TD_1_, TD_2_—orthogonal transverse directions; ND—normal direction.

**Figure 7 materials-13-02041-f007:**
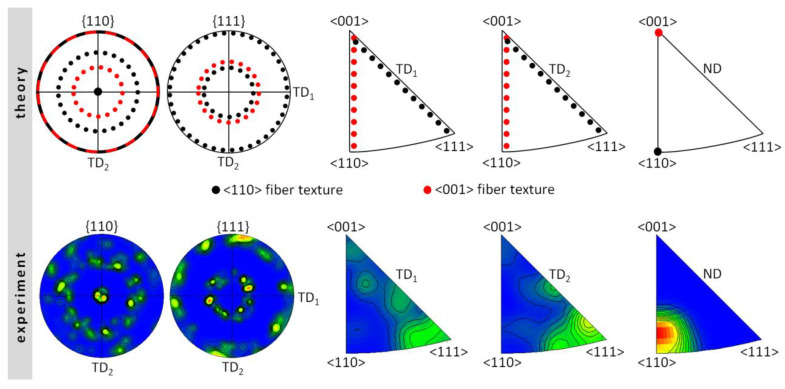
Theoretical and experimental pole figures (PFs) and inverse pole figures (IPFs) for the FeAl alloy after hot extrusion: TD_1_, TD_2_—orthogonal transverse directions; ND—normal direction.

**Figure 8 materials-13-02041-f008:**
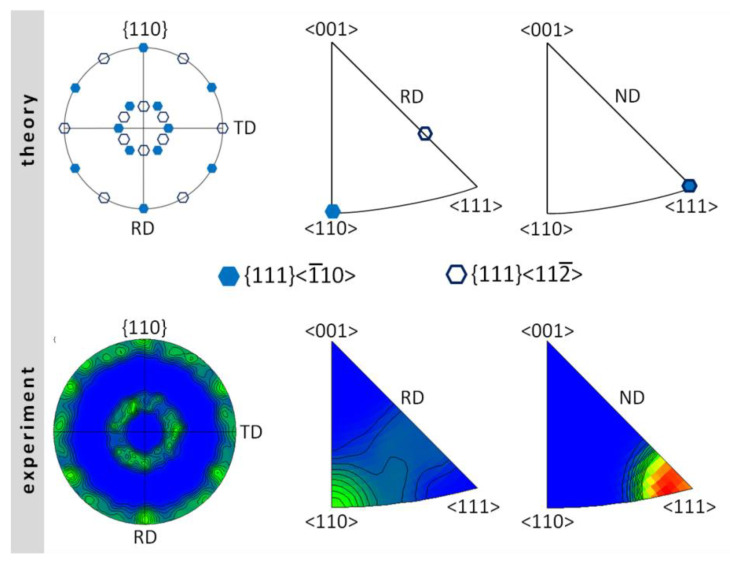
Theoretical and experimental pole figures (PFs) and inverse pole figures (IPFs) for the FeAl alloy after hot rolling: RD—rolling direction, TD—transverse direction, ND—normal direction.

**Figure 9 materials-13-02041-f009:**
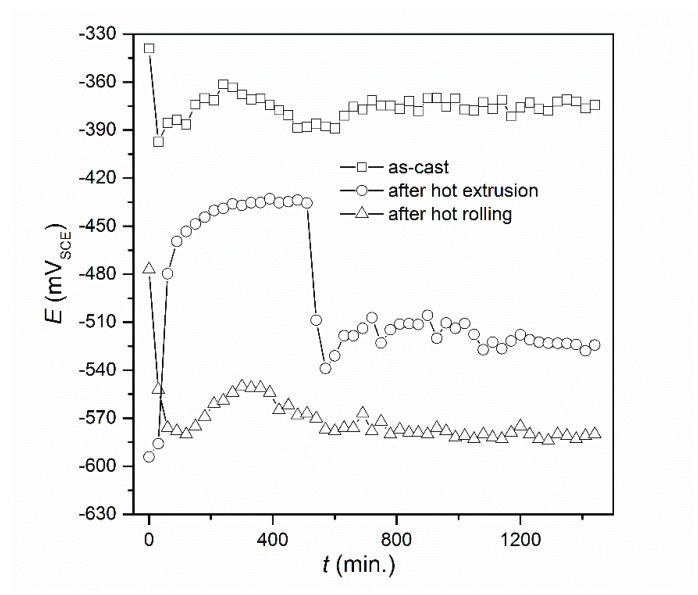
Dependences of E = f(t) for the FeAl alloys.

**Figure 10 materials-13-02041-f010:**
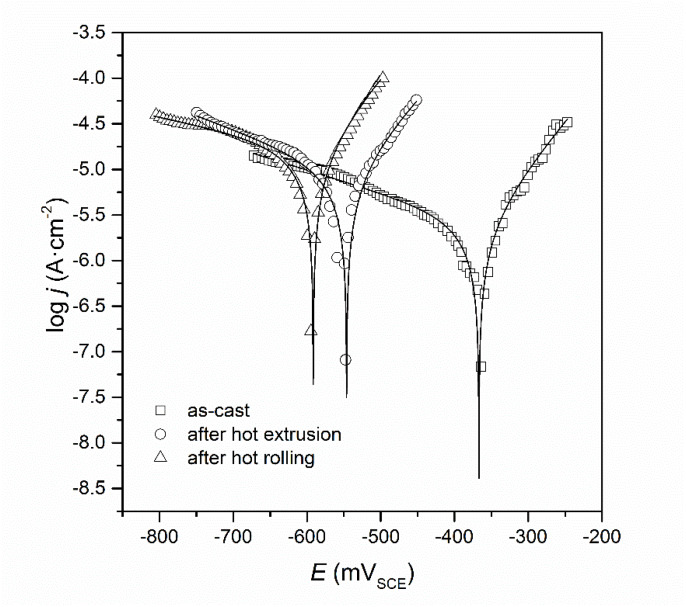
Dependences of log *j* = f(*E*) for the FeAl alloys.

**Figure 11 materials-13-02041-f011:**
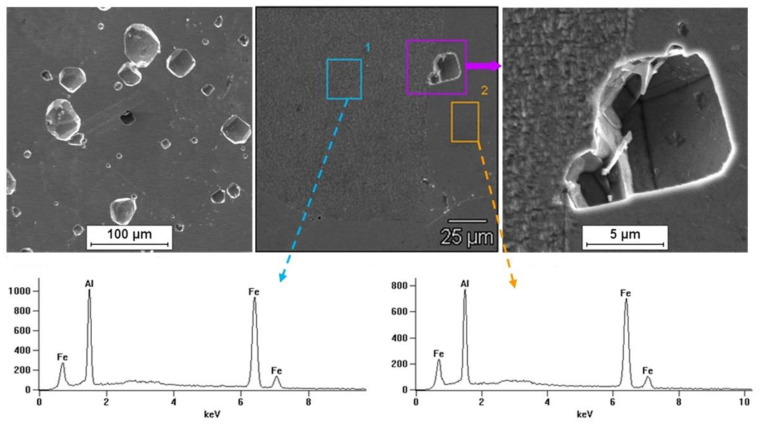
Surface of alloy Fe40Al5Cr0.2TiB after casting and corrosion test.

**Figure 12 materials-13-02041-f012:**
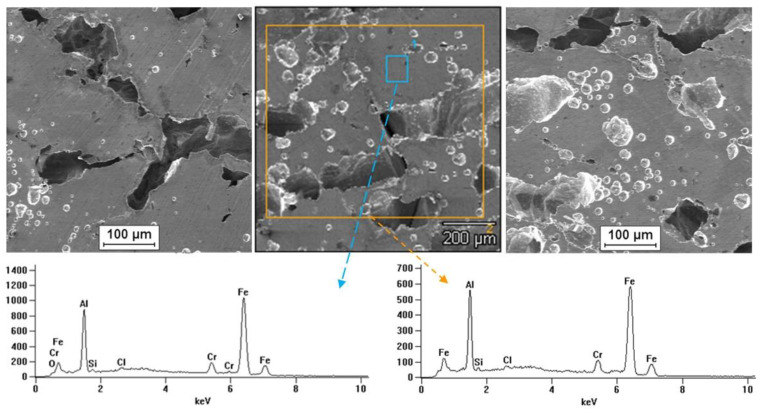
Surface of alloy Fe40Al5Cr0.2TiB after hot extrusion and corrosion test.

**Figure 13 materials-13-02041-f013:**
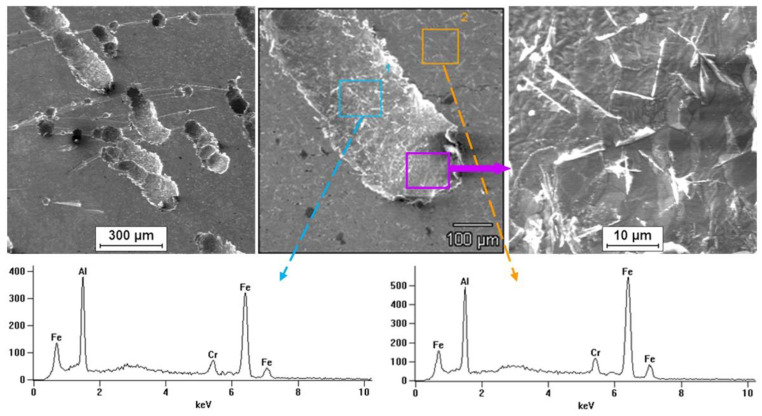
Surface of alloy Fe40Al5Cr0.2TiB after hot rolling and corrosion test.

**Table 1 materials-13-02041-t001:** Chemical composition of Fe40Al5Cr0.2TiB alloy.

Element	Al	Cr	Ti	B	Fe
% mas.	24.53	5.80	0.19	0.01	ball
% at.	40.10	4.86	0.18	0.06	ball

**Table 2 materials-13-02041-t002:** Degree of alloy processing Fe40Al5Cr0.2TiB after plastic processing (d_0_—initial diameter; d_k_—final diameter; S_0_—initial cross-sectional area; S_k_—final cross-section area).

Parameters	After Extrusion	After Rolling
Dimension before Plastic Processing	d_0_ = 22 mm/S_0_ = 380 mm^2^	d_0_ = 25 mm/S_0_ = 491 mm^2^
Dimension after Plastic Processing	d_k_ = 12 mm/S_k_ = 113 mm^2^	d_k_ = 7 mm/S_k_ = 39 mm^2^
Degree of Processing	70%	92%

**Table 3 materials-13-02041-t003:** Grain parameters of the FeAl alloy.

Type of FeAl Alloys	Average Grain Size (mm^2^)	Minimal Grain Size (mm^2^)	Maximal Grain Size (mm^2^)	Standard Deviation (mm^2^)	Variation Coefficient (%)
as-Cast	0.0697	0.00040	0.861	0.146	209.5
after Hot Extrusion	0.0811	0.00040	1.142	0.197	242.9
after Hot Rolling	0.0014	0.00002	0.068	0.003	214.3

**Table 4 materials-13-02041-t004:** Corrosion parameters of all tested FeAl alloys.

Type of FeAl Alloys	*E*_corr_ (mV)	*j*_corr_ (μA·cm^−2^)	*b_a_* (V·dec^−1^)	*b_c_* (V·dec^−1^)	*R_p_* (Ω·cm^2^)	*V*_corr_ (mm·yr^−1^)
as-Cast	−367	2.6	0.103	0.399	13846	0.028
after Hot Extrusion	−546	9.4	0.117	0.329	3836	0.104
after Hot Rolling	−589	16.4	0.112	0.581	2500	0.182

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
