# Peer review of "Evaluation of Structure and Corrosion Behavior of FeAl Alloy after Crystallization, Hot Extrusion and Hot Rolling"

_materials, 2020, doi:10.3390/ma13092041_

Round 1
Reviewer 1 Report
The paper presents the corrosion behavior of Fe40Al5Cr0.2TiB alloy after casting, plastic working using extrusion and rolling methods.
The paper is “honest”: not high research interest is present, maybe probably because the authors do not show completely the importance of this characterization. Considering this point the introduction need an improvement where the important aspects and scientific innovative one are highlighted.
The number on the left is the line number.
35: pay attention to apex of units
61: table 1: instead of “rest” better “ball.”
69: introduce more details about extrusion process (parameters, temperature etc)
In "methods" part introduce more details about “texture analysis” methodology;
About corrosion caracterisation introduce the evaluated samples area.
81: in caption of table 2 (or in text) introduce the corresponded names for d0, dk, S0, Sk
129 and fig.s 1 and 3: after extrusion process deformed grains in direction of extrusion could be expected. Considering the data only a grain reduction is observed. The orientation appears from the misorientation distribution in fig.s 3 and 4; however, it is not easy to observe considering the microstructure pictures. Comments and maybe an more details of this concept in the text are necessary. I am proposing also to improve the clearness of text connected with fig.s 2-4
Fig.s 2, 3 and 4: introduce a marker; in addition, introduce more details about the use of different colors
Fig.s 3 and 4: introduce the direction of working action
Fig. 5: considering that the grain dimension is in the range 0 – 0.05 mm it is necessary to improve details in this range to observe the real distribution of gran dimension. Mandatory.
190: the rate of polarization is just indicated in the “methods” part.
About corrosion characterization:
this is the weakest part of the paper.
Fig.9: introduce comments and interpretation about the quickly changes of potential values for “after extrusion” sample
Table 4: introduce information of the method used to individuate the Vcorr (used experimental data, Tafel slopes, Rp ….)
202: the interpretation of corrosion data is totally not sufficient. The authors report only the description of data and not an interpretation: why the hot rolled sample presents a worse corrosion behavior? The corrosion morphology is uniform or localized; what is the influence of the presence of Cl- ions? The change of behavior is in anodic or cathodic or both parts? ……
The orientation of samples (in direction of extrusion of hot rolling action, perpendicular ….) could be observed differences in the behavior? What the influence of grain borders: there are an intensification of corrosion attack or not?
This part need (mandatory) an improve. On the contrary is not enough.
Ref.s 4, 6 , 7 , 8, 11 substitute with re.fs in English or international journal. Mandatory
Author Response
Response to Reviewer 1
Dear Reviewer. Great thanks for your positive approach and very valuable comments. We tried to improve the text according to your suggestions and we hope that it will satisfy you. All the text changes are cited after your specific comment and highlighted in the text with red colour.
35: pay attention to apex of units
We edited error and changed to: g·cm-3
61: table 1: instead of “rest” better “ball.”
It has been corrected according to suggestion
69: introduce more details about extrusion process (parameters, temperature etc)
We think, that:
The alloys based on the intermetallic phase belong to the group of difficult-to-cut materials. The low plasticity of these alloys requires the use of unconventional plastic working methods. The results of the tests showed changes in the plastic deformation mechanism at high temperatures as indicated by the stress-strain curves. One of the most important parameters of the plastic working process of Fe40Al5Cro, 2TiB alloy is temperature. Lowering the temperature of the deformation process causes a significant increase in the forming resistance and material cracking, which practically prevents the forming of alloys by plastic working. Maintenance of stringent technological conditions causes many difficulties, which as a result of many years of research and tests have been eliminated and are the subject of patent No. 208310. Therefore, the process parameters such as temperature, deformation speed and recipient's geometry are covered by patent protection.
And we modified the introduction
In "methods" part introduce more details about “texture analysis” methodology.
We added necessary information into the section. We added following text:
The analysis of grain size, grain orientation and texture was performed. Orientation maps were displayed in the inverse pole figures color scheme (which allows the crystallographic orientation to be quickly interpreted in terms of the sample coordinate system). Texture analysis was carried out on the basis of pole figures PFs (revealing how plane normals are arranged relative to the specimen) and the inverse pole figures IPFs (revealing which crystallographic directions align with the specimen axes).
About corrosion characterization introduce the evaluated samples area .
We added following information:
The surface apperance of the samples after corrosion tests was studied by Scanning Electron Microscopy (SEM) Hitachi S-4200. The chemical composition was tested using an X-ray energy dispersion spectrometer (EDS) from ThermoNoran (System Seven) at a voltage accelerating the electron beam of 15keV. Spectrometer is connected to the cited microscope.
81: in caption of table 2 (or in text) introduce the corresponded names for d0, dk, S0, Sk
We added in caption of Table 2: (d0 – initial diameter; dk – final diameter; S0 –initial cross-sectional area; Sk – final cross-section area).
129 and fig.s 1 and 3: after extrusion process deformed grains in direction of extrusion could be expected. Considering the data only a grain reduction is observed. The orientation appears from the misorientation distribution in fig.s 3 and 4; however, it is not easy to observe considering the microstructure pictures. Comments and maybe an more details of this concept in the text are necessary. I am proposing also to improve the clearness of text connected with fig.s 2-4
Thank you for your detailed comment. We think that:
The degree of plastic processing as a result of extrusion for hard-deformable materials on the matrix of the FeAl intermetallic phase is not high, and taking into account the significant differences in the stereological parameters of the material after crystallization, the determination of grain deformation in the direction of extrusion is subject to a large error.
Fig.s 2, 3 and 4: introduce a marker; in addition, introduce more details about the use of different colors.
Thank you for so specific comment and we introduced following information in the text
Colors interpretation according to the stereographic triangle (red <001> direction, green <110> direction, blue <111> direction); TD1, TD2 – orthogonal transverse directions; RD – rolling direction.
Fig.s 3 and 4: introduce the direction of working action –
We included in the picture.
Fig. 5: considering that the grain dimension is in the range 0 – 0.05 mm it is necessary to improve details in this range to observe the real distribution of gran dimension. Mandatory. - -
We included in the picture
190: the rate of polarization is just indicated in the “methods” part.
Thank you for this suggestion and we removed the text from line 190.
Fig.9: introduce comments and interpretation about the quickly changes of potential values for “after extrusion” sample.
Thank you for this remark we just forget to input this information in first version. Now we added text lines 243 – 250.
Table 4: introduce information of the method used to individuate the Vcorr (used experimental data, Tafel slopes, Rp ….)
We introduced necessary information in lines: 257,258 and table 4
202: the interpretation of corrosion data is totally not sufficient. The authors report only the description of data and not an interpretation: why the hot rolled sample presents a worse corrosion behavior? The corrosion morphology is uniform or localized; what is the influence of the presence of Cl- ions? The change of behavior is in anodic or cathodic or both parts?
Thank you for suggestion. We made following changes according to the Reviewer's comments. The interpretation in section 4.3 was rebuild follow the suggestions and morphology section is added (4.4) .
In addition, a new section (4.4) on research using a scanning electron microscope (after corrosion resistance tests) has been written.
The orientation of samples (in direction of extrusion of hot rolling action, perpendicular ….) could be observed differences in the behavior? What the influence of grain borders: there are an intensification of corrosion attack or not?
We think that corrosion develops faster with a smaller grain system due to the larger ratio of grain boundary surfaces to the real grain surface, which increases the contact surface and accelerates corrosion.
Ref.s 4, 6 , 7 , 8, 11 substitute with re.fs in English or international journal.
Reviewer 2 Report
The authors presented a study on the effects of plastic deformation treatments on the micro-structure and corrosion resistance. However, there is barely relation or related discussion between the micro-structure and corrosion performances. The following problems should be addressed before publicaiton:
- More backgroud information should be added in the introduction to show the previous studies on this classic topic.
- The relation between the micro-structure and corrosion performances seems opposite to some previous studies, e.g. 10.1016/j.scriptamat.2010.08.035; 10.5006/1.3462912; More explanation and discussion should be provided.
- There are much content on the micro-structure, but the corrosion test is weaker with just the polarization test. A immersion test is suggested to compare their corrosion behaviors.
Author Response
Response to Reviewer 2 Comments
Dear Reviewer. Great thanks for your positive approach and very valuable comments. We tried to improve the text according to your suggestions and we hope that it will satisfy you. All the text changes are cited after your specific comment and highlighted in the text with red and blue colour.
More backgroud information should be added in the introduction to show the previous studies on this classic topic.
The introduction has been supplemented with the information provided. The content was supplemented with the application possibilities of the FeAl alloy, and was also referred to earlier research. The literature review shows that corrosion resistance tests were most often conducted on materials based on the Fe3Al intermetallic phase. The phase equilibrium system shows that this phase is stable only up to a temperature of about 700 ° C. above this temperature there is a phase change which is not good for high temperature applications. Research published in the works of other authors were conducted on the material after casting. Obtaining a material from intermetallic FeAl alloy wrought requires the use of unconventional methods of plastic working due to the fragility of these materials at room temperature and the complexity and technological difficulties of plastic forming processes at elevated temperature. Studies on the effect of corrosion resistance in the liquid environment for an alloy of the indicated chemical composition and after various plastic working processes have not yet been conducted. The rationale for conducting research in the field of corrosion resistance in a liquid environment of heat-resistant alloys based on a matrix of FeAl intermetallic phase is the fact that structural components operating at high temperature are exposed to gases containing acid anhydrides. While the phenomenon of electrochemical corrosion does not occur during operation at high temperature, when the devices are turned off, and consequently, when the temperature drops to room temperature, the water contained in the air (water vapor) may combine with the anhydrides mentioned, resulting in the formation of water acid or salt solutions.
The relation between the micro-structure and corrosion performances seems opposite to some previous studies, e.g. 10.1016/j.scriptamat.2010.08.035; 10.5006/1.3462912; More explanation and discussion should be provided.
Research carried out as part of the indicated work presents a summary of the impact of grain size on the rate of electrochemical corrosion for various materials and environments. These analyses do not include the presentation of corrosion resistance for the intermetallic alloy Fe40Al5Cr0,2TiB, therefore, in order to determine the correlation between the results for different materials, additional tests and analyzes of their results should be performed for the tested alloy. This is due to the fact that, in particular, the intermetallic alloy whose matrix is the FeAl phase has not been characterized in previous studies in a way that gives grounds for the adoption of representative results from a statistical point of view.
There are much content on the micro-structure, but the corrosion test is weaker with just the polarization test. A immersion test is suggested to compare their corrosion behaviors.
Thank you for suggestion, but it is impossible to realization. Instead, we was added a new section (4.4) on research using a scanning electron microscope (after corrosion resistance tests) has been written.
Reviewer 3 Report
This manuscript investigated the structure and corrosion properties of FeAl alloy after different treatments. Some experimental results including SEM, EBSD and corrosion tests were presented, while the scientific contribution is very limited and quite few analysis were conducted.
- There are many typo and language mistakes in the manuscript, it is better to send for proofreading before submission.
- Most of cited references are mentioned/presented in general, please specify it in details, such as **** et al. carried out experimental studies on ***.
- Very limited number of experimental trials were designed in this work, what are the main reasons behind this?
- Although the corrosion properties were determined, while it is better to give more theoretical analysis and explanations for the results rather than mainly present values.
- Some of the results about the change of microstructure and corrosion resistance are quite obvious, it is better to give more novel findings from the work.
In conclusion, in my opinion, the present manuscript is an experimental report rather than a scientific paper. The scientific contribution of this manuscript was very limited and some of the results are not interesting, more depth of works should be conducted to improve the quality of this paper.
Author Response
Response to Reviewer 3 Comments
Dear Reviewer. Great thanks for your positive approach and very valuable comments. We tried to improve the text according to your suggestions and we hope that it will satisfy you. All the text changes are cited after your specific comment and highlighted in the text with red colour.
There are many typo and language mistakes in the manuscript, it is better to send for proofreading before submission.
We agree with the reviewer, the text will be proofread before submission.
Most of cited references are mentioned/presented in general, please specify it in details, such as **** et al. carried out experimental studies on ***.
Thank you for your comment. All has been included in the publication text
Very limited number of experimental trials were designed in this work, what are the main reasons behind this.
The number of experiments presented in the paper was necessary and sufficient to obtain reliable results.
In accordance with the reviewer's comment, we additionally expanded the text of the article with microscopic analysis of the material after corrosion
Although the corrosion properties were determined, while it is better to give more theoretical analysis and explanations for the results rather than mainly present values.
Theoretical data was added in lines 253-258
Some of the results about the change of microstructure and corrosion resistance are quite obvious, it is better to give more novel findings from the work.
Interesting results after heat treatment have been added in section 4.4
Reviewer 4 Report
Dear Authors
I appreciate your effort in carrying out detailed experiment to characterize the microstructure of as-cast, hot rolled and hot extruded FeAl alloy. However, I have the below concerns
- The characterization result does not properly map back to the corrosion resistance performance in 3.5% NaCl.
- Mechanism on the affect of microstructure on the reduced corrosion rates in the case of as-cast FeAl alloy versus hot rolling and hot extrusion is very clear.
- The title should clearly state the knowledge gap so that the readers are attracted to read this manuscript as soon as they see the title.
Author Response
Response to Reviewer 4 Comments
Dear Reviewer. Great thanks for your positive approach and very valuable comments. We tried to improve the text according to your suggestions and we hope that it will satisfy you. All the text changes are cited after your specific comment and highlighted in the text with red colour.
I appreciate your effort in carrying out detailed experiment to characterize the microstructure of as-cast, hot rolled and hot extruded FeAl alloy. However, I have the below concerns
The characterization result does not properly map back to the corrosion resistance performance in 3.5% NaCl.
Thank you for comment. In our opinion, in a 5% NaCl solution, the results are representative for the purpose of describing the course of the corrosion process.
The title should clearly state the knowledge gap so that the readers are attracted to read this manuscript as soon as they see the title.
Thank you for your valuable suggestion. Title has been changed to:
“Evaluation of structure of FeAl alloy after crystallization, hot extrusion and hot rolling after electrochemical corrosion in 5% NaCl”
Round 2
Reviewer 1 Report
The paper in revised version is for me accept with minor revisions (see following comments point to point)
A better title of paper could be:
Evaluation of structure and corrosion behavior of FeAl alloy after crystallization, hot extrusion and hot rolling
Line 63: saline solution
253: the pitting attack is connected with a passive layer. In the studied samples we don’t observe passivity behavior (see fig 10). Modify the text
284: instead of “pits” it is better “localized attacks”
294: instead of X-ray better EDS (= energy X-ray dispersion spectroscopy)
Write the sentence (o similar with the same concept) “ Corrosion develops faster with a smaller grain system due to the larger ratio of grain boundary surfaces to the real grain surface, which increases the contact surface and accelerates corrosion” also in the discussion part after SEM of corroded surfaces.
Author Response
Response to Reviewer 1
Dear Reviewer. Great thanks for your positive approach and very valuable comments. We tried to improve the text according to your suggestions and we hope that it will satisfy you. All comments were corrected in accordance with the Reviewer's recommendations and marked in blue in the original document.
A better title of paper could be:
Evaluation of structure and corrosion behavior of FeAl alloy after crystallization, hot extrusion and hot rolling
Lines 2-4: Title was changed
Line 63: saline solution
Corrected
Line 253: the pitting attack is connected with a passive layer. In the studied samples we don’t observe passivity behavior (see fig 10). Modify the text
Line 248 – 250 The sentence was corrected following the suggestion
„In the case of alloy after hot extrusion increase in the direction of more positive potentials was observed, which at first might have suggested the creation of passive protective layers, however, after eight hours there was a quite rapid decrease leading to stabilization of potential”
Line 284: instead of “pits” it is better “localized attacks”
Line 280: Corrected to “localized attacks”
Line 294: instead of X-ray better EDS (= energy X-ray dispersion spectroscopy)
Line 291: Corrected to EDS
Write the sentence (o similar with the same concept) “ Corrosion develops faster with a smaller grain system due to the larger ratio of grain boundary surfaces to the real grain surface, which increases the contact surface and accelerates corrosion” also in the discussion part after SEM of corroded surfaces.
Lines 293 – 301: Sentence was ccorrected in accordance with the Reviewer's comments
Reviewer 2 Report
The revised manuscript has been improved but there were still two important problems required to be addressed:
1. There is still no detailed explanation and discussion provided for the corrosion behavior. It is not accepted to use just one sentence in the conclusion section as the discussion on this important question in the whole manuscript. The related questions includes: a. How did the corrosion reactions proceed on the alloy surface? b. How about the reactions at the grain boundary? c. What is the differences of the boundary and the matrix? d. How did the difference induce the corrosion preference?
2. For the immersion test, why is it "impossible to realization"? As I know, the immersion test only require the weight change before and after samples socked in the related media.
Author Response
Response to Reviewer 2
Dear Reviewer. Great thanks for your positive approach and very valuable comments. We tried to improve the text according to your suggestions and we hope that it will satisfy you. All comments were corrected in accordance with the Reviewer's recommendations and marked in blue in the original document.
- There is still no detailed explanation and discussion provided for the corrosion behavior. It is not accepted to use just one sentence in the conclusion section as the discussion on this important question in the whole manuscript. The related questions include: a. How did the corrosion reactions proceed on the alloy surface? b. How about the reactions at the grain boundary? c. What are the differences of the boundary and the matrix? d. How did the difference induce the corrosion preference?
Lines 293–301: Sentence corrected in accordance with the Reviewer's comments
- For the immersion test, why is it "impossible to realization"? As I know, the immersion test only requires the weight change before and after samples socked in the related media.
The tests proposed by the Reviewer were not included in the cycle of tests carried out on the prepared material, therefore the surface of the material in subsequent studies was destroyed. This makes it impossible to carry out the experiment. The Fe40Al5Cr0.2TiB intermetallic alloy is characterized by a differentiation of the structure depending on the manufacturing conditions to a greater extent than conventional engineering materials. For example, obtaining an alloy with a small number of metallurgical defects requires re-melting several times. In connection with the above, the production of another material for testing will result in obtaining an alloy whose comparison with the previously tested may be unreliable due to factors that depend on process parameters that are not measurable and statistically variable. For this reason, in the studies presented as part of the work, the analyzes were carried out on material from one melt, which was tested in the state after crystallization, extrusion and rolling.
Reviewer 3 Report
Frankly speaking, although some revisions have made, the technical level of the paper is still below average. However, the authors have tried their best and showed their sincerity. Thus, I give the green light to accept this paper.
Author Response
We thank the expert reviewer for his/her comments
Round 3
Reviewer 2 Report
Accept in the current form.